# The Impact of Environmental Transmission and Epidemiological Features on the Geographical Translocation of Highly Pathogenic Avian Influenza Virus

**DOI:** 10.3390/ijerph16111890

**Published:** 2019-05-28

**Authors:** Xueying Li, Bing Xu, Jeffrey Shaman

**Affiliations:** 1Ministry of Education Key Laboratory for Earth System Modelling, Department of Earth System Science, Tsinghua, Beijing 100084, China; lixueyin14@mails.tsinghua.edu.cn (X.L.); bingxu@tsinghua.edu.cn (B.X.); 2State Key Laboratory of Remote Sensing Science, College of Global Change and Earth System Science, Beijing Normal University, Beijing 100875, China; 3Department of Environmental Health Sciences, Columbia University, New York, NY 10032, USA

**Keywords:** avian influenza, dynamic model, wild bird migration, environmental transmission

## Abstract

The factors affecting the transmission and geographic translocation of avian influenza viruses (AIVs) within wild migratory bird populations remain inadequately understood. In a previous study, we found that environmental transmission had little impact on AIV translocation in a model of a single migratory bird population. In order to simulate virus transmission and translocation more realistically, here we expanded this model system to include two migratory bird flocks. We simulated AIV transmission and translocation while varying four core properties: 1) Contact transmission rate; 2) infection recovery rate; 3) infection-induced mortality rate; and 4) migration recovery rate; and three environmental transmission properties: 1) Virion persistence; 2) exposure rate; and 3) re-scaled environmental infectiousness; as well as the time lag in the migration schedule of the two flocks. We found that environmental exposure rate had a significant impact on virus translocation in the two-flock model. Further, certain epidemiological features (i.e., low infection recovery rate, low mortality rate, and high migration transmission rate) in both flocks strongly affected the likelihood of virus translocation. Our results further identified the pathobiological features supporting AIV intercontinental dissemination risk.

## 1. Introduction

Highly pathogenic avian influenza (HPAI) viruses have repeatedly caused emergent outbreaks in poultry and pose a serious threat to both the poultry industry and human health. Since its first emergence in China in 1996, HPAI H5N1 has caused multiple outbreaks in poultry, and more than 10 million domestic birds have been killed by infection or culled to control the virus [1]. H5N1 infection has also been documented in hundreds of people with a case fatality rate of more than 50% [2,3]. In addition, spillover to wild birds has facilitated the spread of the virus from East Asia to Central Asia, Europe, and Africa in three epidemic phases from 2003 to 2006, and the virus has become endemic in Egypt and Southeast Asia [4,5,6,7]. An ongoing concern is that a human pandemic might arise if the H5N1 virus mutates or reassorts and gains the capacity for sustained human-to-human transmission [8]. 

In 2014, another novel reassortant virus, HPAI H5N8, emerged from the H5N1 Gs/GD lineage and caused large outbreaks in East Asia [9]. The virus spread intercontinentally to Europe, North America and other regions within a single year, and subsequently reassorted with local avian influenza viruses (AIVs) to form new subtypes, including H5N2, H5N3, and H5N6 [10,11,12,13]. H5N8 was the first HPAI virus since 2005 to spread across the northern hemisphere, and the first Eurasian HPAI virus to spread to North America [12]. Given the quick and large-scale dissemination of HPAI H5N8, it is important to identify the attributes that facilitated this rapid geographic translocation. 

Previous research has shown that live-poultry trade and wild bird migration are critical means for spreading HPAI viruses. Experimental studies have indicated that these viruses are not pathogenic to domestic or wild ducks, which is a pre-condition for virus transmission among ducks [14]. HPAIVs are thought to originate from poultry and to move across locations via the transport of infected poultry or contaminated materials [10,15,16,17]. For wild birds, HPAI H5 virus antibodies have been detected in multiple wild duck species in Asia and Europe, which indicates that these birds survive HPAI infection. Additionally, the timing of HPAI virus detection at different geographic locations corresponds to the seasonal migration and flyway paths of wild birds, and there is high spatial correlation between the evolutionary flow of H5N1 and wild bird migration routes in Asia [18,19]. 

A number of studies have analyzed the intercontinental dispersal of H5N8 and shown that it is mainly attributable to the long-distance migration of wild birds. For example, international poultry trade data from the Food and Agriculture Organization are inconsistent with the pattern of H5N8 virus spread from Asia [10]. In contrast, detailed phylogenetic and phylogeographic analyses showed that the virus mainly spread along two long distance migration routes: 1) From the east coast of Asia to Europe; and 2) from the Korean peninsula, across the Bering Strait and into North America [11,20,21,22]. Further research using ancestral host inference and waterfowl habitats and flyways suggested that the virus had been introduced into North America by long-distance migrants [12]. However, it remains unclear why the H5N8 virus spread faster and farther than other HPAI subtypes (e.g., H5N1, H5N6).

In a previous study, we used a compartmental model with environmental transmission to simulate the transmission and geographic translocation of an AIV within one migratory bird population [23]. We found that the virus possessing a lower infection recovery rate had a higher likelihood of translocating to the birds’ summer breeding grounds. Other properties, including contact transmission rate, infection-induced mortality rate, and migration recovery rate, all significantly affected the probability of the virus reaching the summer breeding grounds. Other research has shown that environmental reservoirs help AIVs persist in a particular location, infect more susceptible birds, and even cause secondary outbreaks [24,25]. However, environmental transmission—which has a force of transmission, on average, several orders of magnitude smaller than that of the contact transmission rates—had little impact on AIV translocation in our single-flock model [23]. 

In this paper, we modified our previous compartmental model by adding a second susceptible flock and again included environmental transmission in the model. We used this model framework to study AIV translocation from bird wintering grounds to summer breeding grounds within two flocks as a function of epidemiological properties, environmental parameters, and the time lag in the migration schedule of the two flocks. Specifically, we varied four core epidemiological properties: 1) Contact transmission rate; 2) infection recovery rate; 3) infection-induced mortality rate; and 4) migration recovery rate; and three environmental transmission properties: 1) Virion persistence; 2) exposure rate; and 3) re-scaled environmental infectiousness. The results further describe how environmental transmission affects AIV translocation and identify the features that support HPAI intercontinental spread.

## 2. Method

### 2.1. The Model

We developed a compartment model based on the form presented in Galsworthy et al. [26] and Li et al. [23]. The model simulates the transmission and translocation of AIV in two wild duck flocks (around 5000 individuals in each flock). One flock is seeded with an AIV (Flock A); the other flock is fully susceptible (Flock B). We considered eight distinct geographic patches along a migration path used by both flocks (Figure 1A): One patch for the wintering ground (Patch 1), three for the stopover during spring migration (Patches 2 to 4), one for the summer breeding ground (Patch 5) and three for the stopover during fall migration (Patches 6 to 8). 

Virus transmission occurs through direct contact within the flock and through environmental reservoirs. For each patch and each flock, virus transmission was simulated using a susceptible-infectious-recovered (SIR)-type model (Equation (2a)–(2k)). Because an infection-induced migration delay was considered for birds, the infected (*I*) and recovered (*R*) compartments were divided into $I_{1}$, $I_{2}$ and $R_{1}$, $R_{2}$, respectively. In this model, once infected, birds moved to *I*_1_ and were unable to migrate. Infected birds in *I*_1_ could either recover to *R*_1_ and subsequently regain migratory ability upon transition to *R*_2_, or first regain migratory ability while still infectious and move to *I*_2_, then subsequently recover to *R*_2_ (Figure 1B). Birds in the $I_{1}$ and $R_{1}$ compartments were not healthy enough to migrate but could move to $I_{2}$ and $R_{2}$ with a migration recovery rate (*v*). Birds recovered from infection—from $I_{1}$ to $R_{1}$, or from $I_{2}$ to $R_{2}$—at an infection recovery rate (*r*). Note that the migration recovery rate and the infection recovery rate are independent of one another. We included a mortality rate induced by infection in the $I_{1}$ compartment. A natural mortality rate $\mu_{n}$ was included for all patches, and a hunting mortality rate $\mu_{h}$ was represented in Patch 1 (the wintering ground) and Patches 6 to 8 (the fall migration stopover sites). We also included a fixed birth rate, b, of 40 new birds per day at the summer breeding ground. We did not consider immunity loss in the model. All parameters are shown and described in Table 1.

In addition to contact transmission, infected birds shed the virus into the environment, i.e., water environments, where the virus can persist. In this form, the virion population in the environment, *V*, is affected by two processes:(1)$$\frac{dV}{dt}=\omega I-\eta V$$
where *I* is the number of infected individuals, ω is the virus shedding rate, and η is the virus decay rate in the environment. Dividing Equation (1) by ω yields Equation (2k), where E=V/ω and c=η. The environmental transmission rate is given by $\gamma \left(1-e^{-\sigma V}\right)$, derived from Breban et al. [24]. The constant exposure rate, *γ*, represents the virus consumption rate scaled by water body volume. *σ* is a constant rate related to the empirically determined ID_50_. For ease of calculation, we adjusted the environmental transmission rate into the equivalent form, $\gamma \left(1-e^{-\left(\sigma \omega \right)\left(\frac{V}{\omega }\right)}\right)$*,* and defined *α* = *σω*, the rescaled environmental infectiousness, and E=V/ω, which yielded a re-scaled environmental transmission rate of $\gamma \left(1-e^{-\alpha E_{i}\left(t\right)}\right)$.

Based on the description above, for each patch i=1, 2, …8, the model had five equations for Flock A and B respectively and one equation for virions in the environment: (2a)$$\frac{dS_{A,i}\left(t\right)}{dt}=-\beta_{i}S_{A,i}\left(I_{1,A,i}+I_{2,A,i}\right)-\gamma \left(1-e^{-\alpha E_{i}\left(t\right)}\right)S_{A,i}\left(t\right)-\left(m_{i}+\mu_{n}+\mu_{h,i}\right)S_{A,i}+m_{i-1}S_{A,i-1}+b_{i}$$
(2b)$$\frac{dI_{1,A,i}\left(t\right)}{dt}=\beta_{i}S_{A,i}\left(I_{1,A,i}+I_{2,A,i}\right)+\gamma \left(1-e^{-\alpha E_{i}\left(t\right)}\right)S_{A,i}\left(t\right)-\left(v_{A}+r_{A}+\mu_{d,A}+\mu_{n}+\mu_{h,i}\right)I_{1,A,i}$$
(2c)$$\frac{dI_{2,A,i}\left(t\right)}{dt}=v_{A}I_{1,A,i}-\left(r_{A}+m_{i}+\mu_{n}+\mu_{h,i}\right)I_{2,A,i}+m_{i-1}I_{2,A,i-1}$$
(2d)$$\frac{dR_{1,A,i}\left(t\right)}{dt}=r_{A}I_{1,A,i}-\left(v_{A}+\mu_{n}+\mu_{h,i}\right)R_{1,A,i}$$
(2e)$$\frac{dR_{2,A,i}\left(t\right)}{dt}=v_{A}R_{1,A,i}+r_{A}I_{2,A,i}-\left(m_{i}+\mu_{n}+\mu_{h,i}\right)R_{2,A,i}+m_{i-1}R_{2,A,i-1}$$
(2f)$$\frac{dS_{B,i}\left(t\right)}{dt}=-\beta_{i}S_{B,i}\left(I_{1,B,i}+I_{2,B,i}\right)-\gamma \left(1-e^{-\alpha E_{i}\left(t\right)}\right)S_{B,i}\left(t\right)-\left(m_{i}+\mu_{n}+\mu_{h,i}\right)S_{B,i}+m_{i-1}S_{B,i-1}+b_{i}$$
(2g)$$\frac{dI_{1,B,i}\left(t\right)}{dt}=\beta_{i}S_{B,i}\left(I_{1,B,i}+I_{2,B,i}\right)+\gamma \left(1-e^{-\alpha E_{i}\left(t\right)}\right)S_{B,i}\left(t\right)-\left(v_{B}+r_{B}+\mu_{d,B}+\mu_{n}+\mu_{h,i}\right)I_{1,B,i}$$
(2h)$$\frac{dI_{2,B,i}\left(t\right)}{dt}=v_{B}I_{1,B,i}-\left(r_{B}+m_{i}+\mu_{n}+\mu_{h,i}\right)I_{2,B,i}+m_{i-1}I_{2,B,i-1}$$
(2i)$$\frac{dR_{1,B,i}\left(t\right)}{dt}=r_{b}I_{1,B,i}-\left(v_{B}+\mu_{n}+\mu_{h,i}\right)R_{1,B,i}$$
(2j)$$\frac{dR_{2,B,i}\left(t\right)}{dt}=v_{B}R_{1,B,i}+r_{B}I_{2,B,i}-\left(m_{i}+\mu_{n}+\mu_{h,i}\right)R_{2,B,i}+m_{i-1}R_{2,B,i-1}$$
(2k)$$\frac{dE_{i}\left(t\right)}{dt}=I_{1,A,i}\left(t\right)+I_{2,A,i}\left(t\right)+I_{1,B,i}\left(t\right)+I_{2,B,i}\left(t\right)-cE_{i}\left(t\right)$$

The migration of birds was simulated as a step function $m_{i}\left(t\right)$ [26], with Flock A migrating according to the schedule shown in Figure 1A. The migration rate of birds in Flock A was $m_{A,i}\left(t\right)=0$, if those birds were prescribed to be at Patch *i* at time *t*, and $m_{A,i}\left(t\right)=1$ if the birds were not. For Flock B, we defined a time lag (τ) so that Flock B began migration τ days after Flock A. If birds in Flock A were prescribed to stay at Patch *i* at time *t*, the migration rate of Flock B was $m_{B,i}\left(t+\tau \right)=0$, and $m_{B,i}\left(t+\tau \right)=1$ if birds in Flock A left Patch *i* at time *t*. Birds capable of migration moved to the next patch once reaching the stopover interval. In contrast, birds not healthy enough to fly remained at their current patch until recovering their ability to fly. Based on Galsworthy et al. [27], birds remained at stopover patches during migration for at least 20 days, stayed at the wintering patch for at least three months, and spent the rest of the year at the breeding patch. 

### 2.2. Simulations and Analysis

To simulate viruses with different epidemiological features, we varied four core parameters for each of the flocks: Contact transmission rate ($\beta_{A}$, $\beta_{B}$), infection recovery rate ($r_{A}$, $r_{B}$), infection-induced mortality ($\mu_{d,A}$, $\mu_{d,B}$), and migration recovery rate ($v_{A}$, $v_{B}$). We also varied three environmental parameters: Persistence of virions (*c*); exposure rate (γ); and re-scaled environmental infectiousness (α); as well as the time lag (*t*) between migration start time of the two flocks. The parameter ranges used are shown in Table 1.

We randomly selected 10,000 parameter combinations using Latin hypercube sampling (LHS) based on the ranges given in Table 1. The model was implemented in R (version 3.5.2) and integrated with a time step of 0.5 days for 365 days. Influenza virus was seeded into the simulation on the first day in Flock A, and the initial state of the model is presented in Table 2. To account for stochastic effects, 100 stochastic simulations with each combination of parameters were run. 

To measure the likelihood of virus survival and translocation, for each patch and combination of parameters, we calculated the fraction of the 100 stochastic simulations for which the virus was present, or, conversely, extinct, in a bird flock at a given patch. We then examined all the simulations to determine the combination of pathobiological features favoring virus persistence and geographic dispersal, or, conversely, virus extinction and geographic isolation.

## 3. Results

We focus on the presence and extinction of virus during spring migration (Patches 3 and 5), as Patch 5—the summer breeding ground—is where bird populations from different regions would potentially mingle. We assumed that a virus with a higher likelihood of reaching Patch 5 was more likely to disperse intercontinentally. Figure 2 shows the average likelihood of virus translocation—by either flock—to Patches 3 and 5 as a function of each of the four core parameters in Flock A (the seeded flock) and Flock B (the susceptible flock). For each parameter combination, the likelihood of reaching Patches 3 and 5 is quantified as the fraction of 100 stochastic simulations for which the virus does not go extinct before reaching the patch. 

Sensitivity was found for each of the four parameters in both flocks with the parameters in Flock A exhibiting a greater impact. Among the four parameters, infection recovery rate exhibited the strongest effect. The infection recovery rate in Flock A significantly determined the likelihood that the virus reached Patches 3 and 5, with a lower infection recovery rate strongly increasing the likelihood of virus translocation. The virus reached Patch 3 with a 0.6 or higher likelihood for parameter combinations with an infection period of 10–13 days in Flock A (infection recovery rate of 1/13–1/10 day^−1^). For the parameter combinations with an infection period of 13 days in Flock A, the virus translocated to Patch 5 with a likelihood of approximately 0.6. Patterns could also be seen for the infection recovery rate in Flock B. A lower infection recovery rate in Flock B increased the probability of reaching the summer breeding patch. This effect was more evident when the infection recovery rate in Flock A was higher. Conversely, sensitivity to the infection recovery rate in Flock B was minimal when the infection recovery rate in Flock A was lower. Sensitivities were also apparent for the other three core parameters. The likelihood of reaching Patches 3 and 5 rose as transmission rates in both flocks increased, and a lower mortality rate in both flocks increased the probability of reaching the summer grounds, as did a higher migration recovery rate.

To quantify the sensitivity of virus translocation to each parameter, we computed partial rank correlation coefficients (PRCC) based on the results shown in Figure 2 (Table 3). We found that the absolute value of PRCC for infection recovery rate in Flock A ($r_{A}$) was highest among all parameters, which indicated that Flock A’s infection recovery rate had the strongest impact on virus translocation. The transmission rate and mortality rate in Flock A had absolute PRCC values of about 0.4–0.5, which represented a medium level of impact. The migration recovery rate also had a significant impact on the likelihood of virus translocation, although the PRCC absolute value was the smallest of the four core Flock A parameters. For the Flock B parameters, the PRCC absolute values were smaller but also showed significant impacts on virus translocation.

Figure 3 shows a marginal likelihood that the virus reaches Patches 3 and 5 for each of the three environmental transmission parameters, as well as the migration start time lag between the two flocks. There appears to be little sensitivity to the persistence of virions, environmental infectiousness and time lag; however, for environmental exposure rate, a weak trend exists in which parameter combinations with a higher exposure rate have a higher probability of reaching Patches 3 and 5. We again used PRCC to quantify the sensitivity of the likelihood of virus translocation for the three environmental transmission parameters and time lag (Table 4). Persistence of virions, environmental infectiousness, and time lag all showed PRCC values around 0 and very high p values, whereas environmental exposure rate had a relatively high PRCC absolute value and statistically significant p values smaller than 0.005. This result suggests that the environmental exposure rate significantly impacted the likelihood of the virus reaching Patches 3 and 5, and indicated that environmental transmission had some impact on virus translocation in migrant flocks.

To further investigate the features that always support translocation to a given patch versus never support translocation during spring migration, we plotted the two-dimensional kernel density for each of the four parameters (Figure 4). Always supporting translocation occurred when 100% of the 100 stochastic runs with a particular parameter combination translocated the virus to a given patch, and never supporting translocation occurred when 0% of the stochastic runs translocated the virus to a given patch. We identified all parameter combinations always supporting translocation as well as never supporting translocation and plotted the two-dimensional kernel density for each pair of parameters using an axis-aligned bivariate normal kernel [31] from Package ggplot2 (https://cran.r-project.org/web/packages/ggplot2/index.html) in R (version 2.5.2). 

We found that the likelihood of always translocating to Patches 3 and 5 increased as Flock A infection recovery rate decreased. In addition, there was a decreasing likelihood of virus never translocating as Flock A infection recovery rate decreased. In contrast, Flock B infection recovery rate only had a slight impact on the likelihood of both always translocating and never translocating. The patterns for mortality rate and migration recovery rate were quite similar. The likelihood of always translocating increased with a lower mortality rate in Flock A, as well as a higher migration recovery rate. The mortality rates in Flock A and Flock B combined affected the likelihood of never translocating—there was a decreasing likelihood of the virus never translocating as the mortality rate in both flocks decreased. An increase in the migration recovery rate in both flocks also decreased the likelihood of the virus never translocating. For the transmission rate, parameter combinations with lower transmission rates in both flocks supported the virus never translocating. However, a transmission rate in Flock A around $6\times 10^{-5}$ exhibited the highest density of always supporting translocation, a finding distinct from the results presented in Figure 2.

## 4. Discussion

In this study, we expanded a dynamic model of AIV in migratory birds to further explore the impact of environmental transmission and pathobiological features on HPAI translocation. Our results showed that for this two-flock model system, the epidemiological features of Flock A (the initially seeded flock) largely determined the likelihood of the virus translocating to the summer breeding patch; however, the epidemiological features of Flock B (the fully susceptible flock) also appeared to affect the likelihood of overall virus translocation during bird migration. Further, there existed a significant impact of environmental transmission on virus translocation. 

The pathobiological features identified here favoring virus translocation were similar to the findings of our previous study using a single flock model. The present results corroborated that viruses with a lower infection recovery rate and infection-induced mortality rate were more likely to spread intercontinentally. With a lower infection recovery rate, an AIV has more opportunity to spread due to its longer infectious period. Similarly, with a lower mortality rate, more infected birds survive and continue transmitting the virus. The migration recovery rate also had an impact on AIV spread; at lower migration recovery rates, infected birds remained in a migratory patch for longer, which helped to isolate healthy susceptible birds from infected birds, thereby suppressing transmission.

In addition, higher transmission rates increased the overall likelihood of translocating the virus to the summer breeding patch. Here, the greater transmission rate increased the efficiency and speed of virus transmission; however, at the same time, a higher transmission rate decreased the likelihood of always translocating to the summer breeding patch. This difference may be due to depletion of the number of susceptible individuals during spring migration when transmission rates are high, such that for some stochastic runs the virus goes extinct prior to reaching the breeding patch. This effect needs to be further investigated experimentally or through field observation.

Previously, we found that environmental transmission had little impact on the translocation of a virus in a single-flock model; however, with two flocks, the virus shed into the environment by Flock A showed potential to infect the fully susceptible flock (Flock B). Environmental exposure rates largely determined the likelihood that birds in Flock B were infected, which in turn further affected the overall virus translocation rates. 

Due to environmental transmission, the pathobiological features of AIV in Flock B could also significantly affect virus translocation, though the spread of the virus in Flock B, to some extent, appeared to depend conditionally on Flock A. When the conditions in Flock A are more suitable for the development of infection, more virus particles can be released into the environment, which creates a greater likelihood of Flock B infection. Once infected, the virus will transmit within Flock B mainly through direct contact; however, virus translocation appears more dependent on the features in Flock A. If the features in both flocks favor virus translocation (i.e., lower infection recovery rate and lower mortality rate), the virus has a high likelihood of spreading geographically; if the features in Flock A support virus translocation, but not in Flock B, virus translocation is still likely; however, if the features in Flock A do not favor virus translocation, but the features in Flock B do, translocation to the summer breeding patch remains possible but not as probable.

Our research has some limitations and highlights directions for future research. Firstly, our results focus on virus translocation during spring migration, and we assume that the virus with a larger probability of reaching the summer breeding grounds is more likely to spread intercontinentally. However, bird contact behavior at breeding grounds and subsequent migration—which are not included in this study—may also act as important determinants of intercontinental dissemination. Future studies should simulate these processes to more fully identify the ecological and pathobiological factors supporting intercontinental dispersal.

Environment transmission of AIVs is affected by many factors not represented in our model, including habitat temperature, pH and lake volume. Lower temperature and higher pH favor the persistence of viruses in the environment [25,32]. In addition, lake volume modulates the probability of environmental transmission [25]. Considering these environmental factors in future modeling studies may help to better discriminate the impact of environmental reservoirs on AIV translocation.

A number of other factors, including age structure, loss of immunity and cross immunity, will also need to be considered in the future. For example, birds previously infected with certain low pathogenic avian influenza (LPAI) have some immunity to HPAI. Considering these effects in future studies could help improve understanding of how the age structure and immunity in wild populations can affect AIV transmission and dispersal.

## 5. Conclusions

In summary, the results of our 2-flock model indicate that environmental transmission significantly affect virus translocation within migratory populations. We further identified that certain epidemiological features (i.e., low infection recovery rate, low mortality rate, and high migration transmission rate) in both flocks affect the likelihood of virus translocation and further support AIVs intercontinental dissemination risk.

## Figures and Tables

**Figure 1 ijerph-16-01890-f001:**
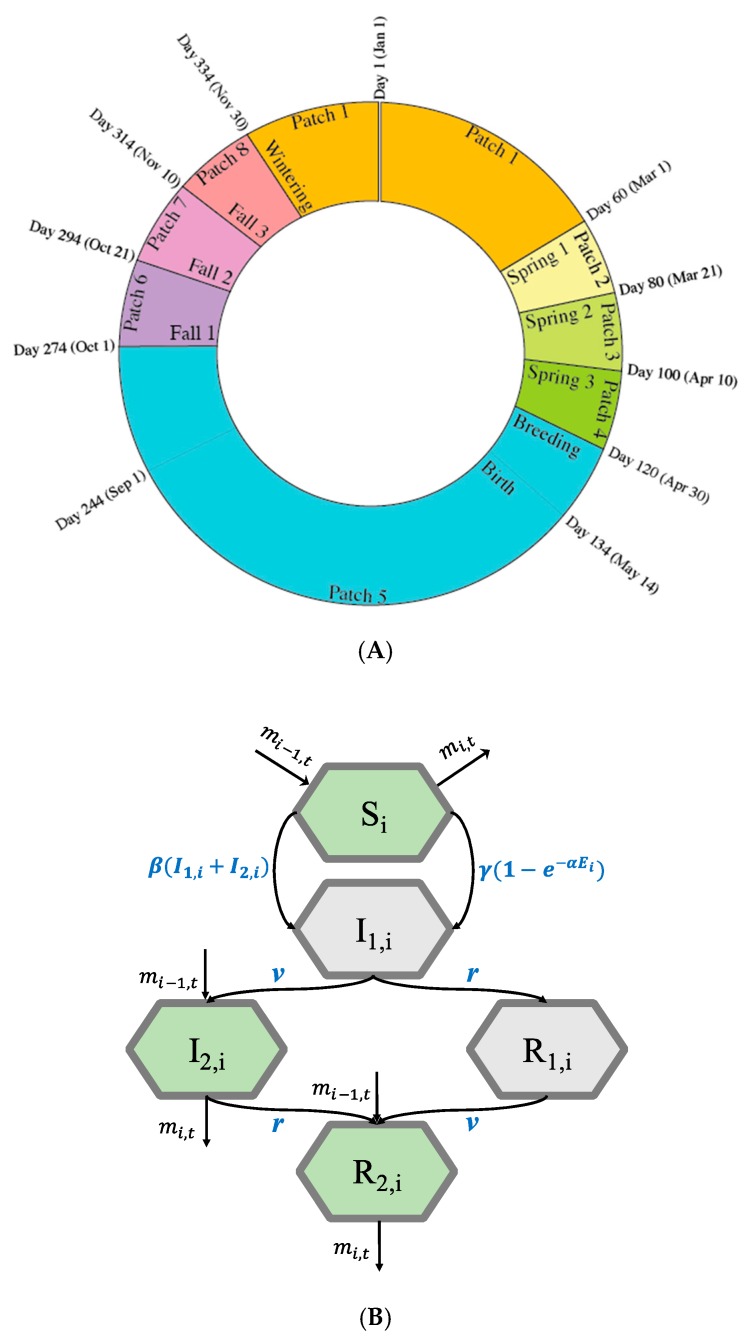
(**A**) Annual migration schedule of the simulated migratory bird populations. The model starts on day 1 and ends on day 365 of a year. The wintering season is from day 334 of the preceding year to day 60 of the next year (Patch 1); (**B**) Model structure showing the movement of birds in between compartments within one flock at Patch *i* [26]. The five compartments are: Susceptible (S), infected and unable to migrate (*I*_1_), infected and able to migrate (*I*_2_), recovered and unable to migrate (*R*_1_), or recovered and able to migrate (*R*_2_). Migration is only possible for birds in compartments S, *I*_2_, and *R*_2_.

**Figure 2 ijerph-16-01890-f002:**
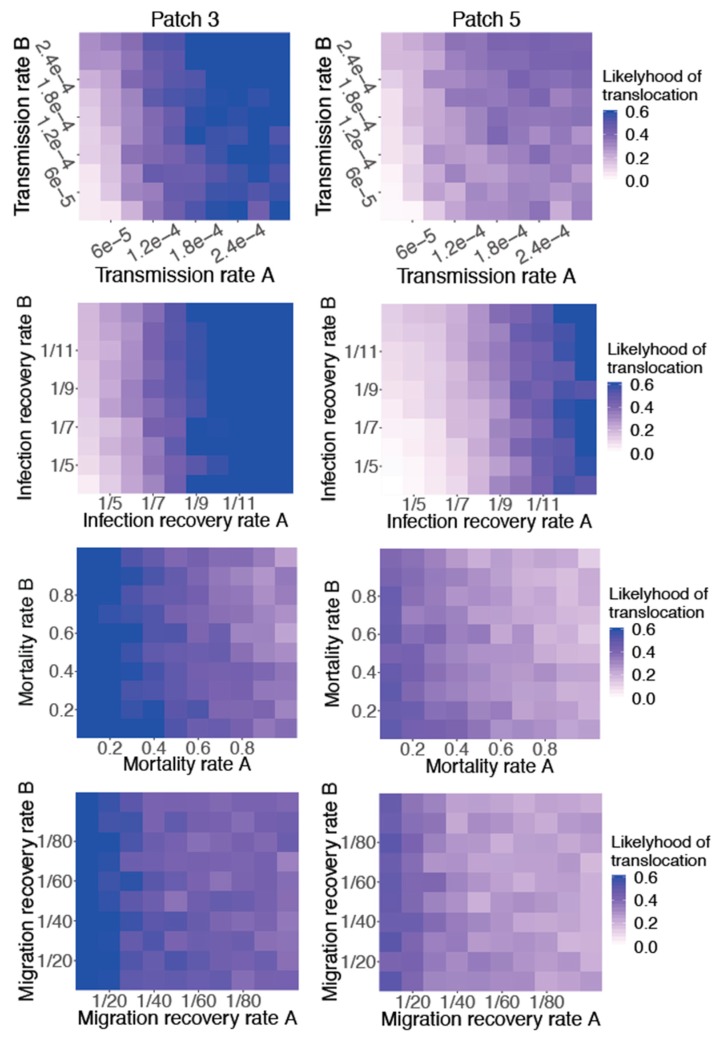
The likelihood of virus translocation to Patch 3 and Patch 5 as a function of the four core parameters in Flock A (the seeded flock) and Flock B (the susceptible flock). The likelihood of reaching Patch 3 or Patch 5 is quantified as the fraction of runs (of 100 stochastic simulations) for which the virus does not go extinct before reaching that patch.

**Figure 3 ijerph-16-01890-f003:**
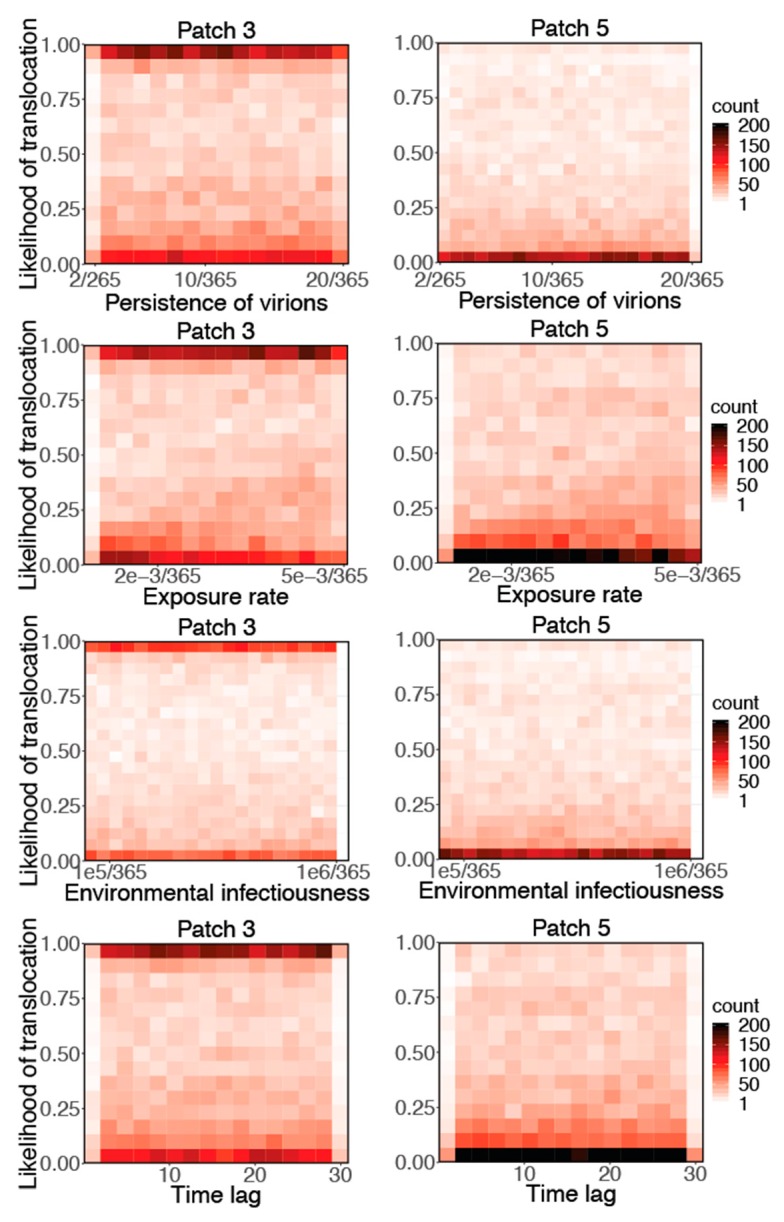
The marginal distribution of the likelihood of the virus reaching Patch 3 and Patch 5 for each of the three environmental transmission parameters: 1) Persistence of virions; 2) exposure rate; and 3) environmental infectiousness; and the migration start time lag between the two flocks. The color gradient represents the number of parameter combinations within each pixel.

**Figure 4 ijerph-16-01890-f004:**
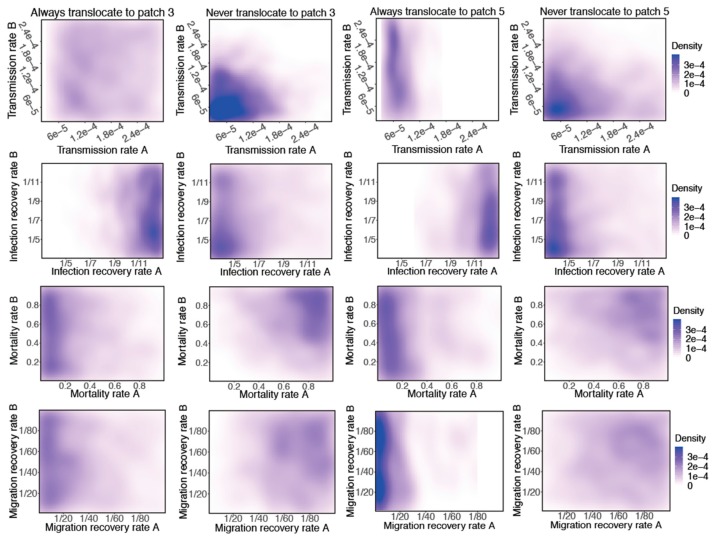
Two-dimensional kernel density for each of the four core parameters in Flocks A and B that always and never support translocating to Patches 3 and 5. A parameter combination always supports translocation if the virus appears in a given patch during 100% of runs with that parameter combination and never supports translocation if the virus appears in a given patch during 0% of runs with that parameter combination.

**Table 1 ijerph-16-01890-t001:** Parameters in the model.

Parameter	Description	Value	Unit	Reference
$\beta_{A,i}$	Contact transmission rate in Flock A	For i≠5, $\beta_{A,i}=b=\left[0.2\times 10^{-4}\;,\;3\times 10^{-4}\right]$; for i=5, $\beta_{A,i}=b/4$	${\mathrm{bird}}^{-1}{\mathrm{day}}^{-1}$	[26,27]
$\beta_{B,i}$	Contact transmission rate in Flock B	For i≠5, $\beta_{B,i}=b=\left[0.2\times 10^{-4}\;,\;3\times 10^{-4}\right]$; for i=5, $\beta_{B,i}=b/4$	${\mathrm{bird}}^{-1}{\mathrm{day}}^{-1}$	[26,27]
$r_{A}$	Infection recovery rate in Flock A	$r_{A}=$ [1/13, 1/3]	${\mathrm{day}}^{-1}$	[26,28]
$r_{B}$	Infection recovery rate in Flock B	$r_{B}=$ [1/13, 1/3]	${\mathrm{day}}^{-1}$	[26,28]
$\mu_{d,A}$	Infection-induced mortality rate in Flock A	$\mu_{d,A}=\left[0,\;1\right]$	${\mathrm{day}}^{-1}$	
$\mu_{d,B}$	Infection-induced mortality rate in Flock B	$\mu_{d,B}=\left[0,\;1\right]$	${\mathrm{day}}^{-1}$	
$v_{A}$	Migration recovery rate in Flock A	$v_{A}=\;$ [1/100, 1/1]	${\mathrm{day}}^{-1}$	[26,29]
$v_{B}$	Migration recovery rate in Flock B	$v_{B}=\;$ [1/100, 1/1]	${\mathrm{day}}^{-1}$	[26,29]
$\mu_{n}$	Natural mortality rate	$\mu_{n}=0.315/365$	${\mathrm{day}}^{-1}$	[26]
$\mu_{h,i}$	Hunting mortality rate	For i=5, $\mu_{h,i}=0.320/365$	${\mathrm{day}}^{-1}$	[26,30]
$m_{A,i}$	Migration matrix of Flock A	Shown in Figure 1.	${\mathrm{day}}^{-1}$	[26]
$m_{B,i}$	Migration matrix of Flock B	Defined by $m_{A,i}$ and τ.	${\mathrm{day}}^{-1}$	[26]
$b_{i,t}$	Birth rate	For i=5 and 162 <t<216, $b_{i,t}=40$, otherwise $b_{i,t}=0$	${\mathrm{birds}\;\mathrm{day}}^{-1}$	[26]
*c*	Persistence of virions	c=[2/365, 20/365]	${\mathrm{day}}^{-1}$	[24,25]
γ	Exposure rate	$\gamma =\left[1\times 10^{-3}/365,\;5\times 10^{-3}/365\right]$	${\mathrm{day}}^{-1}$	[24,25]
α	Re-scaled environmental infectiousness	$\alpha =\left[1/365,\;1\times 10^{6}/365\right]$	${\mathrm{bird}}^{-1}{\mathrm{day}}^{-1}$	[24,25]
τ	The time lag in migration schedule of Flock A and Flock B	τ=[0, 30]	days	

**Table 2 ijerph-16-01890-t002:** The variables of the model.

Variables	Definition	Initial Value
$S_{A}$	Susceptible birds in Flock A	S(0)=4999
$S_{B}$	Susceptible birds in Flock B	$S_{B}\left(0\right)=5000$
$I_{1,A}$	Infectious birds without migration ability in Flock A	$I_{1,A}\left(0\right)=1$
$I_{1,B}$	Infectious birds without migration ability in Flock B	$I_{1,B}\left(0\right)=0$
$I_{2,A}$	Infectious birds with migration ability in Flock A	$I_{2,A}\left(0\right)=0$
$I_{2,B}$	Infectious birds with migration ability in Flock B	$I_{2,B}\left(0\right)=0$
$R_{1,A}$	Recovered birds without migration ability in Flock A	$R_{1,A}\left(0\right)=0$
$R_{1,B}$	Recovered birds without migration ability in Flock B	$R_{1,B}\left(0\right)=0$
$R_{2,A}$	Recovered birds with migration ability in Flock A	$R_{2,A}\left(0\right)=0$
$R_{2,B}$	Recovered birds with migration ability in Flock B	$R_{2,B}\left(0\right)=0$
*E*	Virions in the environment divided by shedding rate (V/ω)	E(0)=1

**Table 3 ijerph-16-01890-t003:** Partial rank correlation coefficient (PRCC) between the likelihood of virus translocating to Patch 3 or Patch 5 and core parameters in both flocks.

Parameter	Patch 3		Patch 5	
	PRCC	*p* Value	PRCC	*p* Value
$\beta_{A}$	0.5133	0.00	0.4149	0.00
$\beta_{B}$	0.2225	0.00	0.2911	0.00
$r_{A}$	−0.6562	0.00	−0.6948	0.00
$r_{B}$	−0.1097	0.00	−0.2305	0.00
$\mu_{d,A}$	−0.4969	0.00	−0.4062	0.00
$\mu_{d,B}$	−0.1478	0.00	−0.1980	0.00
$v_{A}$	0.3595	0.00	0.3255	0.00
$v_{B}$	0.0284	0.00	0.0614	0.00

**Table 4 ijerph-16-01890-t004:** Partial rank correlation coefficient (PRCC) between the likelihood of virus translocating to Patch 3 or Patch 5 and environmental transmission parameters and time lag.

Parameter	Patch 3		Patch 5	
	PRCC	*p* Value	PRCC	*p* Value
*c*	−0.0072	0.47	−0.0063	0.53
γ	0.1201	0.00	0.1437	0.00
α	−0.0060	0.55	−0.0074	0.46
time lag	0.0179	0.07	0.0109	0.28

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
