# Peer review of "The Impact of Environmental Transmission and Epidemiological Features on the Geographical Translocation of Highly Pathogenic Avian Influenza Virus"

_ijerph, 2019, doi:10.3390/ijerph16111890_

Round 1

Reviewer 1 Report

The ideas of the paper is interesting and the results are sound. The paper can be published after some changes. Please see attachment for comments.

Author Response

Response to Reviewer 1’s comments

We thank the reviewer for his/her helpful comments and suggestions. They are incorporated into the revision of the manuscript. Below is a point-by-point response to the reviewer’s specific comments. The original comments are in italics and our response is in normal font.

In this paper, a model system with two migratory bird flocks and environmental transmission is considered. It is reasonable that the factors affecting the transmission and geographic translocation of avian influenza viruses (AIV) within wild migratory bird populations remain inadequately understood. The authors simulate AIV transmission and translocation while varying four core properties and three environmental transmission properties, as well as the time lag in the migration schedule of the two flocks. The ideas of the paper is interesting and the results are sound.

We thank the reviewer for the positive comments.

This paper can be published after some changes. I have the following comments:

1.     The readability of model explanation is low. It is suggested to draw a flow chart or explain it more carefully.

Thank you for your comment and suggestion. The structure of the model in this study is based on the study of Galsworthy et al. (2011). We have added a flow chart of the model (Figure 1B) and more guidance for the model in the revised manuscript (L109 to L112).

2.     In Table 4, you obtain “Environmental exposure rate significantly affects the likelihood of reaching Patch 3 and 5” by comparing the p values of parameters. But please give a detailed explanation in the article.

Thank you for the suggestion. We have added a more detailed explanation to the revised manuscript (L213 to L218).

3.     In paper, it's mentioned several times about "kernel density", but the definition of kernel density is not clear. Please give an explanation.

We have added some description about 2-dimensionial kernel density in the revised manuscript. (L223 to L227)

4.     The formation should be improved. There exist some punctuation mistakes in this paper. For example, the period should come after “(Guan et al. 2004)” on line 41, page 1, and so on.

Thank you for pointing out these mistakes in the manuscript. We have checked and corrected the punctuation mistakes.

5.     Some English grammar mistakes should be corrected.

6.     There are writing errors on Table 1. For example, The word "recovery" should be modified as "mortality" in the description of μd,A and μd,B. There are more similar writing errors on Table 1. The authors should proofread and make the corrections.

We have checked and corrected the grammar mistakes and typos in the manuscript.

7.     Please check the form of the references carefully.

Thank you for your suggestion. We have carefully checked and modified the references as the editor suggested.

Reviewer 2 Report

--

Review comments for manuscript entitled: “The impact of environmental transmission and epidemiological features on the geographical translocation of highly pathogenic avian influenza virus”.

General comments:

Li et al. carried out a mathematical modeling analysis on the factors affecting the transmission and geographic translocation of avian influenza viruses (AIV) within wild migratory bird populations. The current research is based on the extension of their previous study which they found out that environmental transmission had little impact on AIV translocation in a model of a single migratory bird population. In order to simulate virus transmission and translocation more realistically, they have expanded the previous model system to include two migratory bird flocks. They also simulated AIV transmission and translocation while varying four core properties: 1) contact transmission rate; 2) infection recovery rate; 3) infection-induced mortality rate; 4) migration recovery rate; and three environmental transmission properties: 1) virion persistence; 2) exposure rate; and 3) re-scaled environmental infectiousness; as well as the time lag in the migration schedule of the two flocks. They find that environmental exposure rate has a significant impact on virus translocation in the 2-flock model. Furthermore, certain epidemiological features (i.e. low infection recovery rate, low mortality rate, high migration transmission rate) in both flocks strongly affect the likelihood of virus translocation. Their results further identify the pathobiological features supporting AIV intercontinental dissemination risk. The analysis in this paper support the main results with biologically reasonable parameters. This research is certainly interesting and the figures are clear and well explained. Apart from the interesting parts of the work, there are some minor issues that needs revision.

---

Comments

… The authors mentioned that “…bird contact behavior at breeding grounds and subsequent migration, which are not 279 included in this study, may also act as important determinants of intercontinental dissemination. 280 Future studies should simulate these processes to more fully identify the ecological and 281 pathobiological factors supporting intercontinental dispersal….” Why the authors did not considered that in their work as it would reveal more dynamical insights and features of the model.

… Line 36, WHO, 2018 (a) and WHO, 2018 (b) are not listed in the reference list

… Line 67, Torchetti, & Winker, 2015 is not listed in the reference list

--

Typos

... Line 70, “However, it remains unclear why H5N8 virus spread faster”… should be “However, it remains unclear why the H5N8 virus spread faster…”

... Line 76, “…infection induced mortality rate and migration recovery rate …” should be “…infection induced mortality rate, and migration recovery rate …”

… Line 266, “…though the spread of virus in Flock B…” should be “…though the spread of the virus in Flock B…”

… Line 276, “…highlights directions of needed future research…” should be “…highlights directions for needed future research…”

… Line 277, “…we assume that virus with a larger probability…” should be “…we assume that the virus with a larger probability…”

Line 292, “…understanding of how age structure…” should be “…understanding of how the age structure…

Author Response

Response to Reviewer 2’s comments

We thank the reviewer for his/her helpful comments and suggestions. They are incorporated into the revision of the manuscript. Below is a point-by-point response to the reviewer’s specific comments. The original comments are in italics and our response is in normal font.

Li et al. carried out a mathematical modeling analysis on the factors affecting the transmission and geographic translocation of avian influenza viruses (AIV) within wild migratory bird populations. The current research is based on the extension of their previous study which they found out that environmental transmission had little impact on AIV translocation in a model of a single migratory bird population. In order to simulate virus transmission and translocation more realistically, they have expanded the previous model system to include two migratory bird flocks. They also simulated AIV transmission and translocation while varying four core properties: 1) contact transmission rate; 2) infection recovery rate; 3) infection-induced mortality rate; 4) migration recovery rate; and three environmental transmission properties: 1) virion persistence; 2) exposure rate; and 3) re-scaled environmental infectiousness; as well as the time lag in the migration schedule of the two flocks. They find that environmental exposure rate has a significant impact on virus translocation in the 2-flock model. Furthermore, certain epidemiological features (i.e. low infection recovery rate, low mortality rate, high migration transmission rate) in both flocks strongly affect the likelihood of virus translocation. Their results further identify the pathobiological features supporting AIV intercontinental dissemination risk. The analysis in this paper support the main results with biologically reasonable parameters. This research is certainly interesting and the figures are clear and well explained. Apart from the interesting parts of the work, there are some minor issues that needs revision.

We thank the reviewer for the positive comments.

The authors mentioned that “…bird contact behavior at breeding grounds and subsequent migration, which are not 279 included in this study, may also act as important determinants of intercontinental dissemination. 280 Future studies should simulate these processes to more fully identify the ecological and 281 pathobiological factors supporting intercontinental dispersal….” Why the authors did not consider that in their work as it would reveal more dynamical insights and features of the model.

Thank you for this question. The behavior of wild ducks in breeding season is different from that in other periods. Wild ducks usually behave and migrate with the flock. While at their breeding grounds, the ducks will spread out into small groups and different flocks will even mingle together, which means the contact transmission will occur within and between flocks. However, to simulate contact behavior at the breeding grounds in the model, we would first need more field investigation to quantify and thus constrain rates of contact transmission between flocks.

… Line 36, WHO, 2018 (a) and WHO, 2018 (b) are not listed in the reference list

… Line 67, Torchetti, & Winker, 2015 is not listed in the reference list

We have added the corresponding references in the reference list ([2-3,22]).

Typos

... Line 70, “However, it remains unclear why H5N8 virus spread faster”… should be “However, it remains unclear why the H5N8 virus spread faster…”

... Line 76, “…infection induced mortality rate and migration recovery rate …” should be “…infection induced mortality rate, and migration recovery rate …”

… Line 266, “…though the spread of virus in Flock B…” should be “…though the spread of the virus in Flock B…”

… Line 276, “…highlights directions of needed future research…” should be “…highlights directions for needed future research…”

… Line 277, “…we assume that virus with a larger probability…” should be “…we assume that the virus with a larger probability…”

Line 292, “…understanding of how age structure…” should be “…understanding of how the age structure…

We thank the reviewer for helping improve the language in the manuscript. All these typos have been corrected.

Reviewer 3 Report

This study describes a mathematical model that simulates AIV transmission and translocation within and between two migratory bird populations. The question is topical and the findings have the potential to improve our understanding of the ecological factors supporting that support dissemination risk. The paper is very well written, the introduction, the methods and results are very well presented. Some points listed below may be taken into account, and may help to improve the paper.

L98: How did you define these 8 patches? Do you have a reference? Is it the same pattern for all migrating bird species?

L98: Are the two flocks assumed to migrate at the same time or separately in time?

L108: The infection-induced migration delay is not clear. Do you mean that you assumed that birds can be infected and not being able to migrate as they are infected? How do you define the migration recovery rate?

L117: How do you take into account that the flock size might be decreasing if birds are infected and not able to migrate (so they remain in the previous patch compared to the healthy ones which move to the next one)? Please make a link with L132.

L121: How do you define the virus shedding rate? Is it the number of particles of virus excreted over time? The number of infectious doses? How do you get these estimates?

Author Response

Response to Reviewer 3’s comments

We thank the reviewer for his/her helpful comments and suggestions. They are incorporated into the revision of the manuscript. Below is a point-by-point response to the reviewer’s specific comments. The original comments are in italics and our response is in normal font.

This study describes a mathematical model that simulates AIV transmission and translocation within and between two migratory bird populations. The question is topical and the findings have the potential to improve our understanding of the ecological factors supporting that support dissemination risk. The paper is very well written, the introduction, the methods and results are very well presented. Some points listed below may be taken into account, and may help to improve the paper.

We thank the reviewer for the positive comments.

L98: How did you define these 8 patches? Do you have a reference? Is it the same pattern for all migrating bird species?

Thank you for these questions. We developed the model based on the form presented in Galsworthy et al. (2011). In their study, the migration cycle was defined according to a satellite telemetry study of mallards by Yamaguchi et al. (2008). The migration schedules of different species are similar (wintering season, spring migration season, breeding season, and fall migration season), but the detailed migration strategies (number of stopover sites during migration and the time staying at each patch) would be different for different species, even for different populations of same species (Miller et al. 2005; Yamaguchi et al. 2008).

L98: Are the two flocks assumed to migrate at the same time or separately in time?

In the model, we varied the time lag between the migration schedules of two flocks in the model to evaluate the impact of time lag on virus translocation. To be more specific, Flock A (the initially seeded flock) migrates according to the schedule shown in Figure 1. Flock B (the fully susceptible flock) starts to migrate at the same time or after Flock A according to the time lag.  This is described in Lines 134-139 in the revised manuscript.

L108: The infection-induced migration delay is not clear. Do you mean that you assumed that birds can be infected and not being able to migrate as they are infected? How do you define the migration recovery rate?

Thank you for your comment and questions. To better explain the infection-induced migration delay, we present the flow chart of the model in Figure 1B in the revised manuscript.  In this model, once the birds are infected (birds in I1), they are unable to migrate. Infected birds in I1 can recover to R1 but remain unable to migrate, then subsequently regain their migratory ability to R2; or infected birds in I1 can regain the migratory ability to I2 while remaining infectious, then subsequently recover to R2. That is, the infection recovery and migration recovery are independent of each other, and the birds can first recover from infection with infection recovery rate r then regain the ability to migrate with migration recovery rate v, or first regain the ability to migrate with v then recover from infection with r. The migration recovery rate v is defined as the reciprocal of the number of days for birds to regain their migration ability, which is very similar to the infection recovery rate and is calculated as the reciprocal of the infections period.

L117: How do you take into account that the flock size might be decreasing if birds are infected and not able to migrate (so they remain in the previous patch compared to the healthy ones which move to the next one)? Please make a link with L132.

For each patch i=1, 2, ...8, the infection in each flock develops based on the flowchart shown in Figure 1B, which means that we calculate the number of birds in each compartment at each patch for every time step instead of treating each flock as a whole. During integration, birds unable to migrate remain at their current patch (Patch i), and birds with migration ability move on to Patch (L139).

L121: How do you define the virus shedding rate? Is it the number of particles of virus excreted over time? The number of infectious doses? How do you get these estimates?

The virus shedding rate is defined as the number of virions excreted by each duck at each time step per the study of Breban et al. (2009) and it is estimated according to the animal experiments from the study of Webster et al. (1978). The infection dose is used to estimate the rescaled environmental infectiousness (α) in the model. It is defined by α = σω, where ω is the virus shedding rate and σ = loge(2)/ID50, where ID50 is empirically estimated as the dose at which there is a 50% probability of infection) (Breban et al. 2009).

Round 2

Reviewer 1 Report

This article can be accepted.